DATA RELEASE

# Culicidae (Diptera: Culicomorpha) in the southern Brazilian 'Ana Leuch Lozovei' collection, with notes on distribution and diversity

Maurício dos Santos Conceição[1,2,*,†], Samira Chahad-Ehlers[3,†],
Luiz Gonzaga dos Santos-Neto[1], Adson Luís Sant'Ana[1],
Gabriela Princival Marques Ribeiro[1], Débora do Rocio Klisiowicz[1,2],
Cássio Lázaro Silva-Inacio[4,†], Taciano de Moura Barbosa[4],
Renata Antonaci Gama[4], Ana Leuch Lozovei[1] and
Andrey José de Andrade[1,2,5,*]

1 Basic Pathology Department, Federal University of Paraná, Av. Cel. Francisco H. dos Santos, 100 - Jardim das Américas, Curitiba, PR 81531-980, Brazil

2 Postgraduate Program in Microbiology, Parasitology and Pathology, Federal University of Paraná, Av. Cel. Francisco H. dos Santos, 100 - Jardim das Américas, Curitiba, PR 81531-980, Brazil

3 Genetics and Evolution Department, Federal University of São Carlos, Rodovia Washington Luís, km 235 SP-310, São Carlos, SP 13565-905, Brazil

4 Microbiology and Parasitology Department, Federal University of Rio Grande do Norte, Av. Senador Salgado Filho, 3000, Natal, RN 59078-970, Brazil

5 Post-graduate Programme in Entomology, Zoology Department, Federal University of Paraná, Av. Cel. Francisco H. dos Santos, 100 - Jardim das Américas, Curitiba, PR 81531-980, Brazil

**Submitted:** 26 February 2022

\* Corresponding authors. E-mail:
csantos.mauricio@gmail.com;
andreyandrade@ufpr.br

† Contributed equally.

Preprint submitted at https://doi.org/10.1590/SciELOPreprints.3703

Included in the series: *Vectors of human disease* (https://doi.org/10.46471/GIGABYTE_SERIES_0002)

## ABSTRACT

Biological collections are important for acquiring knowledge of the biodiversity of a specific environment. Here, we organize, list and catalog adult specimens of Culicidae from the Parasitology Collection of the Basic Pathology Department at the Federal University of Paraná, Brazil, and present a databank of taxonomic and collection information for each sample. Culicids were collected using different methodologies in 18 municipalities in Paraná state, between 1967 and 1999. There are 5,739 catalogued specimens, of which 4,703 (81.94%) are identified at species level, with a diversity of 100 species. Of these, 18 are new recorded samples for Paraná, and three are new for Brazil. This collection is named the Ana Leuch Lozovei Entomological Collection in honor of the 30 years Prof Lovozei dedicated to culicid study in Paraná. The collection comprises insect vectors or potential vectors, of agents causing human diseases.

**Subjects** Ecology, Biodiversity, Taxonomy

## DATA DESCRIPTION

### Background

The Parasitology Collection of the Basic Pathology Department, housed in the Biological Science Sector of the Federal University of Paraná (ColPar/DPAT/BL/UFPR), Brazil, was established in the 1960s through different research projects. Professor Doctor Ana Leuch Lozovei and her students and collaborators contributed to this initiative by collecting and identifying the different taxonomic groups found in the collection. The collection includes

some insect vector families, notably the family Culicidae, comprising specimens captured in 18 different municipalities in the state of Paraná, in southern Brazil, between 1967 and 1999.

Now retired, Prof Lozovei taught Parasitology and Medical Entomology in the Basic Pathology Department of the Federal University of Paraná. She built collections of diverse taxonomic groups for use in teaching, intending to map species distribution and contribute to advances in medical entomology in Paraná. However, the material gained increasing scientific importance, particularly in medical entomology, ecology, and species identification, organization, diversity and conservation.

To honor the contribution of this researcher, who spent more than 30 years studying the family Culicidae, the Ana Leuch Lozovei Entomological Collection has been created as an integral part of ColPar/DPAT/BL/UFPR. This collection is a valuable legacy that serves as a reference source for researchers interested in the culicid fauna of Paraná.

## Context

The creation of the Ana Leuch Lozovei Entomological Collection is important for several reasons. Firstly, biodiversity is a world heritage and must be valued for the development of each nation. Therefore, as temporary records of organism diversity in an environment, national scientific collections like this one must be seen as the memorial heritage of a country's diversity [1–3]. Secondly, biological collections like this one are important to science, as they enable the development of national strategic research and compliance with international commitments and treaties [1, 4, 5]. Thirdly, they are essential for any researcher who needs to reference a currently recognized name and other information related to any taxon of interest [6, 7].

The epidemiological importance of insects of the family Culicidae (order: Diptera), commonly known as mosquitoes, is another reason why this collection is so valuable. Culicidae have a wide geographic distribution. There are 3,591 currently described species, divided into two subfamilies (Anophelinae and Culicinae), distributed in 113 genera [8]. Around 31% are found in the neotropical region [9, 10]. There are 530 recorded species in Brazil [11], belonging to 23 genera distributed throughout all biomes, and six of these are endemic to the country [12]. There are currently 191 Culicidae species in Paraná [13–15], though as research intensifies, these numbers are increasing [16, 17].

Culicids are also highly relevant to diverse areas of natural and applied sciences, so it is essential to publish and share all known information about these insects with the general and scientific community. Thus, the data collected and catalogued in the Ana Leuch Lozovei Entomological Collection provides a repository of information on the culicid fauna of phytogeographic regions in Paraná in the past. Therefore, the collection has important potential for the utilization of records about species that are currently difficult to collect in the state, contributing to future studies of the fauna and environmental conservation. Most species in the collection are in the subfamily Culicinae, and most specimens (40%) are of the diverse genus *Culex*, distributed among 52 species. By contrast, the genera *Aedeomyia, Lutzia, Onirion* and *Shannoniana* are the least represented, with just one species each. As for Anophelinae, there are two recorded genera, namely *Anopheles*, represented by 30 species, and *Chagasia*, with just one. Having been built over a period of 30 years, the collection provides a timeframe for understanding population fluctuation dynamics, or detection of species not previously recorded, enabling data to be generated in areas of epidemiological potential for diseases related to culicids [18].

These mosquitoes represent an important link in the transmission chain of many neglected, emerging and re-emerging diseases [18–20]. The collection contains samples of diverse species of great medical importance (Table 1 [21–64]); for example, *Aedes* (*Stegomyia*) *albopictus* (Skuse, 1895) and *Aedes* (*Stegomyia*) *aegypti* (Linnaeus, 1762), which are considered the main vectors of dengue (DENV), Zika virus (ZIKV), and Chikungunya (CHIKV). *Ae. aegypti* is also important in the transmission of yellow fever virus (YFV) in the urban environment [65, 66]. *Haemagogus* (*Conopostegus*) *leucocelaenus* (Dyar & Shannon, 1924) and *Ae.* (*Ochlerotatus*) *fulvus* (Wiedemann, 1828), included in the Aedini tribe, are also associated with YFV, but in forested environments [67, 68]. Recent laboratory research [30] observed the vectorial competence of the species *Hg. leucocelaenus* and *Ae.* (*Protomacleaya*) *terrens* (Walker, 1856) for CHIKV, noting that both can transmit this virus. Furthermore, the genus *Culex* is important in the transmission of *Wuchereria bancrofti* (lymphatic filariasis), encephalitis, and serious hemorrhagic fevers such as Oropouche fever [69–71], while the genus *Anophele*s is most responsible for the transmission of *Plasmodium* spp., causing human malaria [72–75].

In addition to their importance for human health, culicids participate in the transmission of diverse pathogens causing diseases of veterinary interest, some of which, such as equine infectious anemia (EIA), dirofilariasis and West Nile fever [76–80], have zoonotic potential, reaffirming the importance of culicids in the epidemiological chain of diseases transmitted by vectors [17, 81, 82].

## METHODS

### Study area

Paraná state is situated in the subtropical region of South America, in southern Brazil, between the coordinates 22° 30′ 44″ S and 26° 43′ 08″ S, and 48° 00′ 11″ W and 54° 36′ 32″ W. It has a territorial area of 199,298 km$^2$ [83], comprising five phytogeographic regions: seasonal semideciduous forest, mixed ombrophilous forest, dense ombrophilous forest, steppe and savanna [84, 85]. According to Köppen classification, the climate in Paraná is divided into subtropical and tropical, with an average annual temperature of 19 °C (varying between 25.9 and 12 °C), and average annual precipitation of 1300 mm [86].

Culicids were captured at nature reserves in 18 municipalities in Paraná between 1967 and 1999: Bela Vista do Paraíso (23° 0′ 0″ S–51° 12′ 0″ W); Campo Mourão (24° 3′ 0″ S–52° 22′ 0″ W); Curitiba (25° 25′ 0″ S–49° 15′ 0″ W); Doutor Camargo (23° 33′ 0″ S–52° 13′ 60″ W); Foz do Iguaçu 25° 33′ 0″ S–54° 34′ 60″ W; Guaíra 24° 4′ 0″ S–54° 15′ 0″ W; Ibiporã 23° 16′ 60″ S–51° 2′ 60″ W; Jataizinho (23° 19′ 0″ S–50° 52′ 60″ W); Londrina (23° 18′ 0″ S–51° 8′ 60″ W); Maringá (23° 25′ 0″ S–51° 55′ 0″ W); Morretes (25° 28′ 31″ S–48° 49′ 45″ W); Ortigueira (24° 12′ 28″ S–50° 54′ 54″ W); Paranaguá (25° 30′ 46″ S–48° 32′ 34″ W); Quatro Barras (25° 22′ 0″ S–49° 4′ 60″ W); Ribeirão Claro (23° 11′ 51″ S–49° 45′ 24″ W); Santa Helena (24° 55′ 60″ S–54° 22′ 60″ W); Santo Antônio do Caiuá (22° 41′ 60″ S–52° 22′0″ W); and Tomazina (23° 46′ 45″ S–49° 57′ 21″ W) (Figure 1). Most of the samples in the collection are from the dense ombrophilous forest phytogeographic region (*n* = 4,831), with 4,774 (98%) of these being from the municipality of Morretes. Bearing in mind the availability of conservation areas in the region, Morretes has long been used by different researchers as a place for culicid collection [87–90].

**Table 1.** Species of mosquitoes (Diptera: Culicidae) present in the state of Paraná, Brazil, and associated pathogens.

| Species | Pathogens | References |
|---|---|---|
| *Aedes aegypti* | Dengue virus (DENV:1,2,3,4); Chikungunya virus (CHIKV); Yellow fever virus (YFV); Zika virus (ZIKV); Venezuelan equine encephalitis virus (VEEV); Mayaro virus (MAYV) | [21, 22] |
| *Aedes albopictus* | Dengue virus (DENV); Chikungunya virus (CHIKV); Yellow fever virus (YFV); Zika virus (ZIKV); Venezuelan equine encephalitis virus (VEEV); Mayaro virus (MAYV) | [21, 22] |
| *Aedes argyrothorax* | Ilheus virus (ILHV); Wyeomyia virus (WYOV); Ilheus virus (ILHV); Yellow fever virus (YFV) | [21, 23] |
| *Aedes fluviatilis* | Chikungunya virus (CHIKV); Yellow fever virus (YFV); *Dirofilaria immitis* | [24, 25] |
| *Aedes fulvithorax* | Yellow fever virus (YFV) | [26] |
| *Aedes fulvus* | Yellow fever virus (YFV); St. Louis encephalitis virus (SLEV); Ilheus virus (ILHV); Western equine encephalitis virus (WEEV) | [21] |
| *Aedes hastatus* | Venezuelan equine encephalitis virus (VEEV); Pixuna virus (PIXV) | [27, 28] |
| *Aedes scupularis* | Ilheus virus (ILHV); Kairi virus (KRIV); Lukini virus (LUKV); Maguari virus (MAGV); Mayaro virus (MAYV); Melao virus (MELV); Oropouche virus (OROV); Playas virus (PLAV); St. Louis encephalitis virus (SLEV); Venezuelan equine encephalitis virus (VEEV); Wyeomyia virus (WYOV); Yellow Fever virus (YFV); *Dirofilaria immitis; Wuchereria bancrofti* | [29] |
| *Aedes serratus* | Aura virus (AURAV); Caraparu virus (CARV); Venezuelan equine encephalitis virus (VEEV); St. Louis encephalitis virus (SLEV); Guama virus (GMAV); Ilheus virus (ILHV); Mirim virus (MIRV); Mucambo virus (MUCV); Oriboca virus (ORIV); Oropouche virus (OROV); Una virus (UNAV) | [21] |
| *Aedes terrens* | Chikungunya virus (CHIKV) | [30] |
| *Aedomyia squamipennis* | *Plasmodium* spp. (aviary); Venezuelan equine encephalitis virus (VEEV); Gamboa virus (GAMV) | [30–33] |
| *Anopheles albimanus* | Chikungunya virus (CHIKV); Sindbis virus (SINV); Semliki Forest virus (SFV); Tlacotalpan virus (TLAV); *Plasmodium berghei; Plasmodium falciparum; Plasmodium vivax* | [34] |
| *Anopheles albitarsis* | Las Maloyas virus (LMV); St. Louis encephalitis virus (SLEV); Western equine encephalitis virus (WEEV); *Plasmodium falciparum; Plasmodium malariae; Plasmodium vivax* | [35] |
| *Anopheles argyritarsis* | *Plasmodium vivax* | [36] |
| *Anopheles bellator* | *Wuchereria bancrofti; Plasmodium* spp. | [25, 37] |
| *Anopheles costai* | *Plasmodium vivax; Plasmodium falciparum* | [38, 39] |
| *Anopheles cruzii* | *Plasmodium vivax; Plasmodium falciparum* | [40] |
| *Anopheles darlingi* | *Plasmodium falciparum; Plasmodium malariae; Plasmodium vivax; Wuchereria bancrofti* | [41] |
| *Anopheles fluminensis* | *Plasmodium brasilianum; Plasmodium falciparum; Plasmodium malariae; Plasmodium simium; Plasmodium vivax* | [42] |
| *Anopheles galvaoi* | *Plasmodium vivax; Plasmodium falciparum* | [43] |
| *Anopheles homunculus* | *Plasmodium falciparum; Plasmodium* spp.; *Plasmodium vivax* | [44] |
| *Anopheles lutzii* | *Plasmodium vivax* | [45] |
| *Anopheles mediopunctatus* | *Plasmodium falciparum* | [46] |
| *Anopheles oswaldoi* | *Plasmodium falciparum; Plasmodium malariae; Plasmodium vivax* | [47] |
| *Anopheles parvus* | *Plasmodium vivax* | [48] |
| *Anopheles rangeli* | *Plasmodium brasilianum; Plasmodium falciparum; Plasmodium malariae; Plasmodium simium; Plasmodium vivax* | [49] |
| *Anopheles rondoni* | *Plasmodium vivax; Plasmodium falciparum* | [50] |
| *Anopheles strodei* | *Plasmodium falciparum; Plasmodium malariae; Plasmodium* spp.; *Plasmodium vivax* | [51] |
| *Anopheles triannulatus* | Breu Branco virus; Las Maloyas virus (LMV); *Plasmodium falciparum; Plasmodium vivax* | [52] |
| *Coquillettidia venezuelensis* | Catu virus (CATUV); Guama virus (GMAV); Itaporanga virus (ITPV); Moju virus (MOJUV); Mucambo virus (MUCV); Oriboca virus (ORIV); Oropouche virus (OROV) | [21] |
| *Culex bidens* | Venezuelan equine encephalitis virus (VEEV) | [53] |
| *Culex coronator* | West Nile virus (WNV); Zika virus (ZIKV); St. Louis encephalitis virus (STLV) | [54] |

**Table 1.** (Continued)

| Species | Pathogens | References |
|---|---|---|
| *Culex declarator* | Oropouche virus (OROV); St. Louis encephalitis virus (SLEV);West Nile virus (WNV) | [55] |
| *Culex nigripalpus* | Cabassou virus (CABV); Eastern equine encephalitis virus (EEEV);Everglades virus (EVEV); Hart Park virus (HPV); Keystone virus (KEYV); St. Louis encephalitis virus (STLV); Tensaw virus (TENV); Venezuelan equine encephalitis virus (VEEV); West Nile virus (WNV); Vesicular stomatitis (NJ serotype) virus (VSNJV); Wyeomyia virus (WYOV) | [56] |
| *Culex ocossa* | Babanki virus (BBKV); Para virus (PARAV); Venezuelan equine encephalitis virus (VEEV); Western equine encephalitis virus (WEEV) | [57] |
| *Haemagogus leucocelaenus* | Yellow fever virus (YFV); Chikungunya virus (CHIKV); Ileus virus (ILHV); Maguari virus (MAGV); Una virus (UNAV); Wyeomyia virus (WYOV) | [21, 30, 58] |
| *Limatus durhamii* | Maguari virus (MAGV); Guama virus (GMAV); Tucunduba virus (TUCV) | [21] |
| *Limatus flavisetosus* | Maguari virus (MAGV) | [21] |
| *Mansonia titillans* | Venezuelan equine encephalitis virus (VEEV); *Dermatobia hominis* eggs | [25, 59] |
| *Psorophora albipes* | Yellow fever virus (YFV); Venezuelan equine encephalitis virus (VEEV); Guama virus (GMAV); Ilheus virus (ILHV); Kairi virus (KRIV); Mayaro virus (MAYV); Una virus (UNAV); Wyeomyia virus (WYOV) | [21] |
| *Psorophora ciliata* | West Nile virus (WNV); Eastern equine encephalitis virus (EEEV); Venezuelan equine encephalitis virus (VEEV); Tansal virus | [60] |
| *Psorophora cingulata* | Oropouche virus (OROV) | [61] |
| *Psorophora confinnis* | Venezuelan Equine Encephalitis virus (VEEV) | [62] |
| *Psorophora ferox* | Guaroa virus (GROV); Ieri virus (IREIV);Ilheus virus (ILHV); Kairi virus (KRIV); Maguari virus (MAGV); Mayaro virus (MAYV); Oriboca virus (ORIV); Oropouche virus (OROV); Una virus (UNAV); Venezuelan equine encephalitis virus (VEEV); West Nile virus (WNV); Wyeomyia virus (WYOV); *Dermatobia hominis* eggs | [25, 63] |
| *Psorophora lutzii* | Guama virus (GMAV); Ilheus virus (ILHV); Una virus (UNAV) | [21] |
| *Sabethes albiprivus* | Yellow fever virus (YFV) | [64] |

## Collecting methods

The geographical coordinates of the 18 sample municipalities were extracted from the platform Global Gazetteer (Version 2.3) [91]. These data were the basis for creating a map covering the phytogeographic areas [85], utilizing *QGIS* software (V. 3.22.3-Białowieża, RRID:SCR_018507) [92]. The culicids in the collection were captured by different methods, such as entomological nets, aspiration, US Centers for Disease Control (CDC) light traps, and Shannon traps containing a fluorescent bulb as bait [13, 87, 88].

## Cataloguing process for the culicids

Cataloguing work was carried out at ColPar/DPAT/BL/UFPR. The classification utilized for Culicidae follows the system proposed by Harbach [8], and Wilkerson *et al.* [9] for the tribe *Aedini*, with abbreviations according to Reinert [93].

Only information on dry preserved male and female culicids, mounted with entomological pins was used for the dataset. Data on individual specimens were tabulated on a spreadsheet using Microsoft Office® v. 2016 (Microsoft Corporation, Redmond, WA, USA) and later transferred to a Darwin Core spreadsheet [94].

In total, the collection contains 5,739 specimens, representing two subfamilies of Culicidae, with seven tribes, 16 genera, 24 subgenera and 100 species. A total of 944

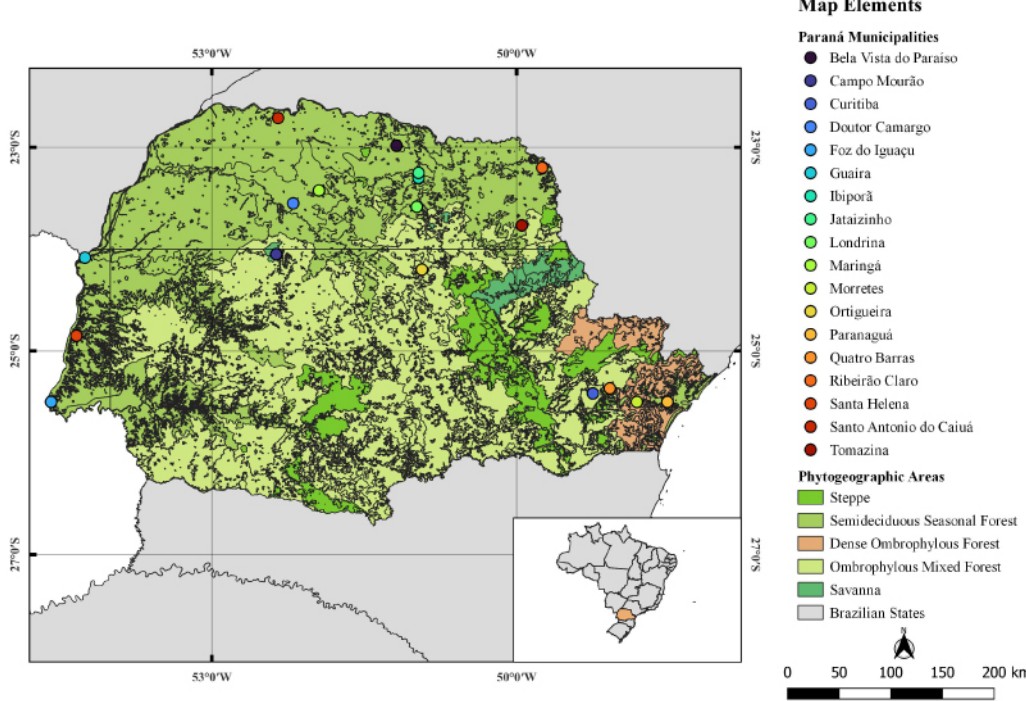

**Figure 1.** Municipalities and phytogeographic regions of the state of Paraná, where the culicid samples housed in the Ana Leuch Lozovei Entomological Collection of the Federal University of Paraná, Brazil were collected.

individuals (16.44%) were identified only at subgenus level and 92 (1.60%) at genus level. Eighteen species constitute the first recorded sample for Paraná state (*Anopheles albimanus* Wiedmann, 1820, *A. costai* da Fonseca & da Silva Ramos, 1940, *A. rangeli* Gabaldon, Cova-Garcia & Lopez, 1940, *Culex aquarius* Strickman, 1990, *Cx. bastagarius* Dyar & Knab, 1906, *Cx. foliaceus* Lane, 1945, *Cx. alinkios* Sallum & Hutchings, 2003, *Cx. faurani* Duret, 1968, *Cx. lucifugus* Komp, 1936, *Cx. hedys* Root, 1927, *Cx. ocossa* Dyar & Knab, 1919, *Cx. oedipus* Root, 1927, *Cx. lanei* Oliveira Coutinho & Forattini, 1962, *Ae. fulvithorax* (Lutz, 1904), *Cx. theobaldi* (Lutz, 1904), *Cx. pleuristriatus* Theobald, 1903, *Ae. eucephalaeus* (Dyar, 1918) and *Uranotaenia ditaenionota* Prado, 1931). Three of these (*Cx. aquarius*, *Cx. lucifugus* and *Ae. eucephalaeus*) are newly recorded samples for Brazil, signifying the expansion of geographical distribution of the species previously restricted to certain locations or countries.

## DATA VALIDATION AND QUALITY CONTROL

Over the years, Dr. Ana Leuch Lozovei and her group have published several works on culicid fauna in Paraná state [13, 73, 88, 89, 95–97], reinforcing the importance of culicids in the transmission of diseases and in environmental quality. Moreover, it was possible to carry out this work through the data obtained. Data validation was also carried out via the GBIF data validator tool upon submission of the data [98].



## REUSE POTENTIAL

Climatic alterations with constant modifications in the environment over time promote significant changes in biodiversity, above all in the insect population, where there is greater dispersion in the environment, mainly in species of medical/epidemiological interest. The data made available in this work have fundamental potential for future studies in ecology, the environment and biodiversity, especially for species of medical interest from the perspective of public health. An important point about the current collection is that it represents 18 first-recorded species for Paraná state, serving as an important reference for research into the biodiversity of mosquitoes. This demonstrates and reaffirms the importance of maintaining and constantly updating biological collections.

## DATA AVAILABILITY

The dataset supporting this article is available in the SiBBr repository [99].

## EDITOR'S NOTE

This paper is part of a series of Data Release articles working with GBIF and supported by the Special Programme for Research and Training in Tropical Diseases (TDR), hosted at the World Health Organization [100].

## DECLARATIONS
## LIST OF ABBREVIATIONS

BL: Biological Sciences Sector; ColPar: Parasitology Collection; DPAT: Basic Pathology Department; TDR: Special Programme for Research and Training in Tropical Diseases; UFPR: Federal University of Paraná.

## ETHICAL APPROVAL

Not applicable.

## COMPETING INTERESTS

The authors declare that they have no competing interests.

## FUNDING

Not applicable.

## AUTHORS' CONTRIBUTIONS

MSC: writing of the manuscript, compilation and organization of data; SCE: sample collecting, identification, data revision and writing of the manuscript; LGSN: sample collecting, identification, data revision; ALSA: sample collecting and identification; GPM: compilation and organization of data; DRK: writing of the manuscript, collection maintenance; TMB: writing of the manuscript; CLSI: writing of the manuscript and data revision; RAG: writing of the manuscript and data revision; ALL: sample collecting, identification; AJA: writing of the manuscript, compilation and organization of data.
All authors read and approved the final version of the manuscript.

## ACKNOWLEDGEMENTS

We would like to thank Clara Baringo Fonseca for her help in preparing the Darwin Core spreadsheet, and Maria Anice Mureb Sallum of the Faculty of Public Health, University of

São Paulo, for identifying some culicid samples in the collection. We would also like to thank the municipalities and institutions for their contributions that enabled sample collection, and Ueslei Teodoro of the State University of Maringá and Mara Cristina Pinto of the State University of São Paulo (UNESP, Araraquara), for their support in the field.

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
