## [Reviewer Report]

Upload additional filesDRR-202202-02/form/DRR-202202-02_Data-Review-CIH.pdfReviewer name and names of any other individual's who aided in reviewer Christopher HunterDo you understand and agree to our policy of having open and named reviews, and having your review included with the published papers. (If no, please inform the editor that you cannot review this manuscript.)YesIs the language of sufficient quality?YesPlease add additional comments on language quality to clarify if needed
Are all data available and do they match the descriptions in the paper? NoAdditional CommentsData are available, but there are a number of differences between the manuscript descriptions and the data as presented in GBIF, please see the additional files PDF for more details.Are the data and metadata consistent with relevant minimum information or reporting standards? See GigaDB checklists for examples <a href="http://gigadb.org/site/guide" target="_blank">http://gigadb.org/site/guide</a>YesAdditional CommentsIs the data acquisition clear, complete and methodologically sound?YesAdditional CommentsIs there sufficient detail in the methods and data-processing steps to allow reproduction?YesAdditional CommentsIs there sufficient data validation and statistical analyses of data quality? YesAdditional CommentsIs the validation suitable for this type of data?YesAdditional CommentsIs there sufficient information for others to reuse this dataset or integrate it with other data?YesAdditional CommentsAny Additional Overall Comments to the AuthorThe manuscript describes the generation of a GBIF dataset from the collection of culicids over a 3 decade period in 18 municipalities in the state of Paraná, largely by Professor Ana Leuch Lozovei together with her research team and students. The collection has been named in her honour.
The paper is well written, but it has been written based on the original dataset as collected with no consideration given to the GBIF dataset generated. Given the tight deadlines expected for this series paper it is understandable that the GBIF upload was perhaps rushed abit, so I hope the authors can now take some time in the revision process to address the known issues flagged in GBIF and make appropriate comments/adjustments to the manuscript to reflect and address differences between the original dataset collected and that which is available via GBIF.

Number of GBIF datasets included in the manuscript - 1

Major comments (Author action required):
1 - In general the manuscript appears to be describing the dataset as it was collected in the host institute, but it does not accurately reflect the data that had been successfully uploaded to GBIF. Where there are differences between the dataset as originally archived in the host institute and the dataset as presented in GBIF these should be highlighted and explained, or where possible, fixed in one or both of the repositories.The GBIF “known issues” information is a good starting point as it highlights those issues that arise from automated processing of the submitted dataset(s).
- Particularly the GPS coordinates need to be corrected in the GBIF dataset
- The number of identified occurrences as the species level is far fewer in the GBIF data than the authors indicate it should be.

Minor comments (Author action suggested):
1 - The URL https://collectory.sibbr.gov.br/collectory/public/show/co446?lang=en_UK has no records in it, it appears to just be a placeholder with minimal metadata. Should there be data available here? If so it should be made public, if not please explain what this URL offers?

2 - Table 1. Species of mosquitoes (Diptera: Culicidae) present in the state of Paraná, Brazil,
and associated pathogens. - 
This table is too large to include as a supplemental file with the manuscript it should be converted to CSV and uploaded to GigaDB.
The table include 203 named species, This is different to the text of the MS stating 205 and the GBIF data containing 99. There needs to be a consistent account of the species identified.

3 - The locality information appears to be messed up in translation to GBIF, there are a couple of dates that are outside the expected range.

Please see the additional PDF for more details.
RecommendationMajor Revision

---

## [Reviewer Report]

Upload additional filesDRR-202202-02/form/Culicid Collection Manuscrit (reviewer comments).docxReviewer name and names of any other individual's who aided in reviewer Olga Lucia Cabrera QuinteroDo you understand and agree to our policy of having open and named reviews, and having your review included with the published papers. (If no, please inform the editor that you cannot review this manuscript.)YesIs the language of sufficient quality?YesPlease add additional comments on language quality to clarify if needed
Are all data available and do they match the descriptions in the paper? YesAdditional CommentsAre the data and metadata consistent with relevant minimum information or reporting standards? See GigaDB checklists for examples <a href="http://gigadb.org/site/guide" target="_blank">http://gigadb.org/site/guide</a>YesAdditional CommentsIs the data acquisition clear, complete and methodologically sound?YesAdditional CommentsIs there sufficient detail in the methods and data-processing steps to allow reproduction?YesAdditional CommentsIs there sufficient data validation and statistical analyses of data quality? YesAdditional CommentsIs the validation suitable for this type of data?YesAdditional CommentsIs there sufficient information for others to reuse this dataset or integrate it with other data?YesAdditional CommentsAny Additional Overall Comments to the AuthorI have reviewed the manuscript and find it adequately described, with just a few suggestions highlighted in the document. Particularly the references and table 1. In two attached documents appear my contributions.RecommendationMinor Revision